# Unveiling Sri Lanka's brain drain and labour market pressure: A study of macroeconomic factors on migration

**Sandunima Kaluarachchi**[1], **Ruwan Jayathilaka**[2]*

**1** SLIIT Business School, Sri Lanka Institute of Information Technology, Malabe, Sri Lanka, **2** Head—Department of Information Management, SLIIT Business School, Sri Lanka Institute of Information Technology, Malabe, Sri Lanka

* ruwan.j@sliit.lk

**Data Availability Statement:** All relevant data are within the manuscript and its Supporting Information files.

## Abstract

The purpose of this study is to explore the impact of GDP per capita income (GDPPCI), unemployment, higher education (HE), and economic growth (EG) on migration in Sri Lanka. Numerous global and local studies have explored the influence of macroeconomic and socioeconomic factors on migration. In the Sri Lankan context, fewer studies have probed the impact of GDPPCI, unemployment, HE, and EG on migration, particularly concerning brain drain and domestic labour market pressure. An applied research methodology was adopted, utilising annual data from 1986 to 2022. The statistical data were sourced from reports by the Sri Lanka Bureau of Foreign Employment (SLBFE), the Central Bank of Sri Lanka (CBSL), Labor Force Survey Data from the Department of Census and Statistics (LFSDCS), and University Grants Commissions (UGC). This study utilised the Vector Error Correlation model (VECM), Vector Auto-regression (VAR), and Granger Causality test through STATA. The empirical findings of the VAR model highlighted that GDPPCI and EG negatively impact migration, whereas unemployment and HE positively affect migration. The study's implications demonstrated that GDPPCI, unemployment, HE, and EG were the primary factors influencing the country's migration decisions. These findings will hopefully inform and guide the Sri Lankan government and policymakers for more effective decision-making.

## Introduction

The issues of labour market pressure and brain drain have become increasingly prominent in developing countries in the twenty-first century. The debate on development and migration has regained prominence in policy circles, specially following the "economic crisis" in Sri Lanka. The country's development is intricately linked to the utilisation of human resources and capital. Hidayat, Onitsuka [1] argued that people are socially dynamic by nature and tend to physically move(migrate) from one place to another for various purposes. Numerous empirical studies validate skepticism towards policies attempting to deter migration through

**Funding:** The author(s) received no specific funding for this work.

**Competing interests:** The authors have declared that no competing interests exist.

development, highlighting their potential ineffectiveness [2, 3]. These skeptical views on macroeconomic and socioeconomic migration factors imply the effect of GDPPCI, unemployment, HE, and EG on migration. Bhardwaj and Sharma [4] showed in 2016 that approximately 3.25% of people in developing countries live outside their country of birth, with employment being one of the primary factors. This figure increased to 3.6% in 2020, representing a substantial migratory population of around 244 million (58%), residing in developed countries, a figure that expanded to 281 million in 2020.

According to the most recent data from SLBFE [5] approximately 311,056 Sri Lankans left the country for foreign employment in 2022, resulting in a monthly departure rate of over 29,000 people. A detailed examination of the data for the same year shows that the number of people leaving the country for professional-level jobs increased by 4.6%, but the number of low-skilled employees significantly increased by 33.92%. Forecasts based on this data suggest a doubling of these statistics in 2023.

Recent economic literature has shown an increased focus on migration trends in developing countries [6]. This analysis assumes that migration is influenced by push factors (stemming from unfavourable conditions in Sri Lanka) and pull factors (associated with favourable conditions in destination countries). In Sri Lanka, factors such as economic instability caused by ethnic conflict, poor educational facilities due to irresolute political governance, displacement, and entrenched authoritarianism significantly contribute to emigration as push factors [3]. The primary pull factors include higher anticipated wages in destination countries, robust social security systems, and political stability [7]. Over the years, the Gulf Cooperation Council (GCC) countries have drawn a substantial influx of foreign labour due to their accelerated economic surge, tax-free status, and high salaries, resulting in a significant domestic labour shortage in various sectors [8]. The scarcity of domestic labour could hamper economic growth in the home country.

Conversely, certain physically demanding and low-paying occupations have demonstrated a lack of favourability among domestic labour forces in Sri Lanka. Therefore, this escalating migration pressure is characterised by an oversupply of individuals seeking to leave the country. This situation triggered a swift wave of skilled and unskilled workers finding job opportunities in foreign countries. As a result, the domestic labour market faced risk, and existing workers were under tremendous work pressure, creating a stressful work environment in Sri Lanka.

Another prominent migration trend is the outflow of young people for foreign education, leading to a major departure from Sri Lanka. This tendency is concerning because it implies the government may not receive a return on its investment in their primary education, thereby unable to leverage their contribution to development in the country. Over the last two decades, there has been consistent growth in the enrollment of international students from developing countries to developed countries, with an 186% increase between 1998–2018 surpassing the 152% growth in the total number of student enrollment [9]. The potential for a high lifetime income is diminished for youth, particularly with a weak economy. Therefore, increased student migration trends align with the host country's job opportunities and economic growth. Gunarathne and Jayasinghe [10] postulate that youth unemployment is more than half the overall unemployment rate; they make migration decisions rather than stay "unemployed" in Sri Lanka. As a result, middle-class populations also give in to student migration in the country [11]. However, from an economic perspective, it is not easy for returning students, having completed their studies, because migration often reflects some students' desire to enter the labour market in the destination country. The study by Rozhenkova [12] highlights a solid inclination to emigrate, particularly among the young and educated populations, exacerbating the country's existing issue of brain drain. The term "brain drain" symbolises the adverse

impact of a loss of a qualified workforce on the country experiencing this talent exodus. While this subject lost prominence in the international debate during the 70s, it has recently regained attention, featuring prominently in the plethora of publications about brain drain [13]. Over time, the emigration of skilled individuals from developing countries like Sri Lanka and India has surged, resulting in a significant brain drain for these countries. Therefore, concern has been raised regarding the outflow of professionals from low-income to more developed countries.

The significance of this study, compared to extant studies, lies in its four-fold emphasis. First, the study examines the influence of GDPPCI, unemployment, HE, and EG on migration in Sri Lanka using data collected over a long period (1986–2022). Only a few prior studies in Sri Lanka have been conducted in this aspect. Hence, this study is unique regarding the extent of Sri Lankan migration data and the years covered. Second, there have been past studies conducted on the determinants of brain drain and migration. However, more arguments need to be made about how macroeconomic and socioeconomic factors impact migration.

Consequently, this study endeavours to improve extant literature with the assistance of previous literature and various techniques such as time-series VECM, VAR, and Granger-causality. Third, recent studies have focused on the critical evaluation of migration and remittances as an option for the economic recovery market and the impact on the economy. Therefore, this study's findings would benefit the country in formulating policies to safeguard the domestic labour market and prevent brain drain. Fourth, the exodus of individuals from the country is intensifying pressure on the domestic labour market, leading to stressful work environments. Notably, the medical and higher education sectors in Sri Lanka are grappling with significant challenges due to a shortage of human capital, emphasizing the need for this study to shed light on the macroeconomic factors influencing migration.

The remaining sections of this article are categorised as follows: Section two explores the relevant literature, while section three outlines the data and methodology. Section four assesses the findings and offers a discussion, while section five provides the research's conclusion.

## Theoretical framework

The neoclassical microeconomic theories identified as the human capital theory of migration, employ cost-benefit analysis and guide individuals to make rational decisions based on the available information available. Expanding beyond individual-level factors, migration theories are often categorised into push and pull theories, and various macro, meso and micro-level web factors influence migration [14]. Push factors play a crucial role in the early stages of the migration decision. Helbling and Morgenstern [15] highlighted that push factors refer to the circumstances, challenges, or disadvantages in the home country that create the impetus for individuals to consider migration. These factors include poverty, income inequality, unemployment, poor economic conditions, and limited access to education and resources. As migrants progress in their decision-making process and move towards planning their migration journey, pull factors become more significant. Pull factors are attractions and opportunities in destination countries that entice migrants, such as employment opportunities, economic prospects, higher wages, educational facilities, healthcare systems, political stability, and safety.

Macro-level influences are factors beyond the control of individuals, households, or communities, such as demographic, economic, and environmental factors. For instance, political instability, conflicts, and a lack of opportunities for career growth professionals may prompt individuals to leave the country. Meso-level factors include diaspora links, migration costs, political and legal frameworks, and the role of employment agencies or migrant smuggling

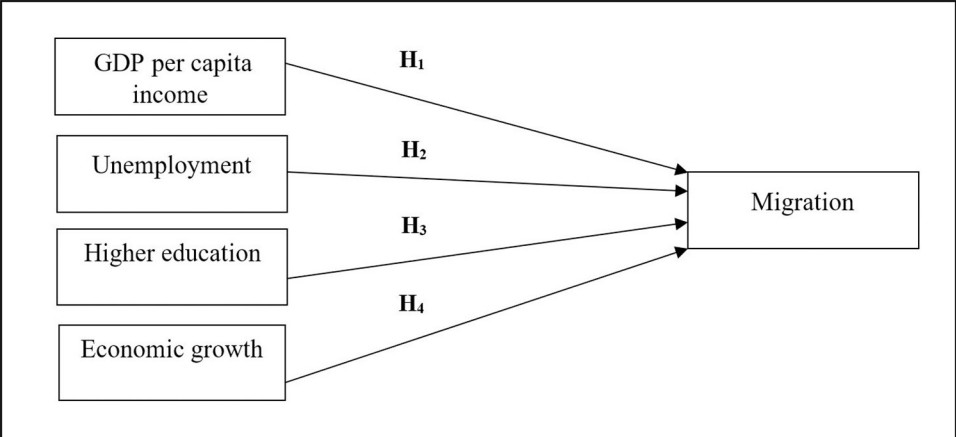

**Fig 1. Conceptual framework.** Source: Authors' compilation.

networks. For example, Negombo and Batticaloa highlight the presence of migrant smuggling networks operating in areas in Sri Lanka, facilitating unauthorized migration to Italy and Australia [14]. Micro-level factors encompass the characteristics of individuals and households, such as health, skills, physical capital, financial capital, social capital, human capital, and the perception of the risk level. Based on household needs, past experiences, future aspirations, social norms, and beliefs, individuals make decisions to migrate.

## Literature review

Building upon the theoretical foundations of migration, the researchers formulated the conceptual framework for the present study, as shown in Fig 1. Sri Lanka is grappling with significant brain drain and labour market strain challenges as its citizens increasingly seek foreign employment opportunities and pursue studies overseas. The study highlights the significance of GDPPCI, unemployment, HE, and EG as critical factors in migration.

### Impact of GDPPCI on migration

The impact of GDPPCI on migration in the context of globalisation has sparked ongoing debates and conflicting views in economic theory and empirical studies. The 2030 Agenda of the Sustainable Development Goals (SDGs) recognises the urgency of addressing income disparities and poverty. While developed countries tackle these challenges, developing countries like Sri Lanka are still processing efforts to enhance living standards, eradicate poverty, and reduce income disparity in line with the SDGs [7, 14, 16]. Countries with lower and medium levels of GDPPCI experience higher emigration rates, while higher-income countries have lower emigration rates [2]. Over the years, Sri Lanka has witnessed a growing trend in migration and, at present, a significant surge in migration in the country due to the current economic crisis. According to IPS [17] the ongoing financial crisis in Sri Lanka stems from a range of factors, including terrorist attacks (Easter Sunday bombings), climate changes, fertiliser issues, natural disasters, and the COVID-19 pandemic. The government's tax hikes and rising costs for essentials like electricity, water, food, etc., have become unbearable for many. For example, certain medications have led to higher out-of-pocket costs for vital medical services. Consequently, individuals opt for migration to respond to the current financial hardships, where GDPPCI plays a pivotal role in these decisions. As per statistics, the majority of departures are recorded by low-skilled employees such as office clerks, construction labours,

housekeepers, security guards, etc., making the domestic labour market profoundly affected by this departure, especially the construction industry.

## Impact of unemployment on migration

Moreover, people's decisions to remain in their home country or move to another destination are often influenced by their income or job. Macroeconomic evidence suggests that employment status is a significant factor, as unemployed individuals are more likely to migrate, and emigration rates are higher in countries with a high unemployment rate [3]. It is noted that the link between unemployment and migration does not always align with economic logic [18], and some wealthier segments may experience more joblessness. At the same time, poorer households rely on all family members to contribute to income [19]. According to the International Labor Organization (ILO), unemployment refers to individuals without jobs who have actively sought employment within the past four weeks [10, 20, 21]. The influencing factor for unemployment is "asymmetric capabilities" within the labour market, where a mismatch between employer demands and job seeker qualifications makes it difficult for candidates to find suitable employment in Sri Lanka.

On the other hand, IPS [17] underlines that the Sri Lankan unemployment rate increased by 0.7% during the COVID-19 pandemic. Consequently, IOM [22] explains that foreign job opportunities, especially in GCC countries, have attracted South Asian workers, helping to reduce domestic unemployment. Thus, rising unemployment rates and the challenge of securing well-paying jobs that alleviate poverty further influence migration decisions. Previously, the government showed limited concern for the unemployment issue. Therefore, after prolonged waiting for long time periods to find suitable jobs, these peoples are leaving the country feeling with frustration. Nevertheless, in the present situation, highly qualified and experienced peoples are emigrating, leading to a shortage of suitable candidates for organizational positions.

## Impact of HE on migration

Furthermore, education is transformative in advancing economic growth and social progress aligned with the SDGs [23]. The research conducted by Chau, Bana [24] pursuing a university degree consistently links with higher lifetime earnings, broader professional networks, and increased career adaptability. The current trend is that Sri Lankan students seek foreign education and leave the country. IPS [25] reveals students face difficulty completing degrees due to limited state university capacity, political interference, campus violence, frequent strikes, and university closures. Degraded HE systems in Sri Lanka are a primary "push factor" prompting individuals to seek education abroad. Existing literature shows that international students are highly likely to remain in the host country for employment after completing their studies. The claim is Zhan, Downey [26] that various factors that could influence international students' decisions to stay in host countries for employment are affected by individual-level, higher educational institutions (HEI), and country-level factors.

Similarly, developed countries attract and retain these international students. Vosko [27] showed Canada's post-graduation work permit program (PGWPP) as a pivotal element for enabling students to work in Canada after completing their studies. Currently, 48% of international students from developing countries in this program are permanent residents in Canada. In 2018, Sri Lanka witnessed 3,750 resident visas issued for students and scholarship holders [28]. Based on past studies evidence, those who migrated for HE had a history of prior unemployment in developing countries [9, 29]. As a result, the literature review reveals that HE impacts migration, including both pull and push factors. This trend directly impacts the brain

drain in the country. Because after finishing their HE, the majority of students often acquire citizenship or permanent residency. Therefore, the native country cannot take these students' talent to develop the country. For example, doctors and lecturers after finishing their education, enter the labour force in their destination country for huge salaries or other benefits. Because they feel that if they return to their home country they cannot earn the money at least invested in their HE.

In the case of many destination countries, wages for specific occupations significantly exceed those in Sri Lanka. IPS [25] statistics reported in 2013, the average annual earning of a qualified CIMA member in Sri Lanka was US$ 22,038, while a similarly qualified CIMA member earned US$ 193,465 in the US, US$ 54,355 in the UK, US$ 135,409 in Australia and US$ 55,955 in the UAE. Also, in 2015 a medical officer in Sri Lanka was entitled to an annual salary of approximately US$ 9,212 (RS. 1.3 million), whereas the same position earned US$ 159,000 in the UK, US$ 84,000 in Australia and US$ 161,000 in the US. Therefore, the allure of better economic prospects and improved living standards makes most students enter the labour force in the destination country, thereby creating brain drain and labour market pressure in Sri Lanka.

## Impact of EG on migration

Moreover, studies found evidence that EG may affect migration in developing countries [30]. Sri Lanka is grappling with a severe economic and political crisis, marked by mounting debt, sovereign default, restricted credit access, and worsened by the economic impact of the COVID-19 pandemic. The country's EG sharply declined from 2.3% in 2019 to -7.8% in 2022 [31]. Furthermore, the current financial crisis has profoundly impacted Sri Lankan's economy, leading to business closures, increased income inequality and substantial job and income losses. In July 2019, Sri Lanka became an upper-middle-income country (UMIC), with a GNI per capita of USD 4,060 [19]. Nonetheless, as a result of the current financial crisis, it reverted to a low-middle-income country (LMIC) in 2023.

Consequently IPS [32], the government views encouraging labour migration as a vital strategy to enhance remittance inflow, aiming to generate essential foreign exchange and alleviate the financial challenges. Islam [33] provides evidence for personal remittances that have several positive impacts on developing economics. As a result, in 2022, low-skilled workers (34%), skilled workers (30%) and housemaids (25%), Kuwait (79,123), Qatar (71,954), Saudi Arabia (53,702) and UAE (35,563) hosted the highest numbers of Sri Lankan employees [5]. Nevertheless, Lanati and Thiele [34] findings reveal a lower migration rate of low-income individuals to the Organisation for Economic Co-operation and Development (OECD) countries, resulting in lower remittances in these destinations, primarily due to the comparatively higher cost of living than GCC countries. Chambers, Bliss [35] reported that the desire for improved economic prospects and better living standards drives many people to seek opportunities abroad, reflecting the concept of "place utility". Hence, EG has a significant influence on migration in Sri Lanka. The government's focus on boosting the EG primarily through increased remittances overlooks the pressing issues of business closures, unemployment, income inequality, and poverty in Sri Lanka. Furthermore, the decision to raise taxes without concurrent improvements in employee salaries and benefits adds stress and frustration to the populace. Therefore, the government's actions increased social unrest and migration trend.

As shown in Fig 1, the conceptual framework was developed with the literature review and existing knowledge. Four hypotheses were combined in the development of this model. These independent variables have been identified as a significant influence on migration decisions.

**Table 1. Variables and supporting studies.**

| Variable | Past studies |
|---|---|
| GDPPCI $\longrightarrow$ Migration (-) | Restelli [2], Bhardwaj and Sharma [4], Mohamed Aslam and Alibuhtto [36], Bollyky, Graetz [37], Ahmad and Arjumand [38], Islam and Khan [39] |
| Unemployment $\longrightarrow$ Migration (+) | Redlin [3], Gunarathne and Jayasinghe [10], Bonifazi, Heins [18], Atigala, Maduwanthi [20], Gardiyawasam, Ganegoda [21] |
| HE $\longrightarrow$ Migration (+) | Weber and Van Mol [9], Rozhenkova [12], Zhan, Downey [26], Vosko [27], Rehák and Eriksson [29], Khan and Bin [40] |
| EG $\longrightarrow$ Migration (-) | Rasamoelison, Averett [30], Islam [33], Lanati and Thiele [34], Khan and Arokkiaraj [41], Tipayalai [42] |

Source: Authors' compilations.

Table 1 categorises the supporting literature into four independent variables related to the study's migration: GDPPCI, unemployment, HE, and EG.

## Data and methodology

A theory addressing the effects of GDPPCI, unemployment, HE, and EG on migration was tested using the adopted applied research methodology, which incorporates quantitative and secondary data collection approaches. The study justifies the selection of data and the chosen method by referencing relevant research articles. The summary of the methodological characteristics of each is provided in S1 Appendix. Building on the strong support from previous studies for the selected methodology, the present study also adopts the same approach.

The VAR approach was employed to analyse the data. This study chose these models due to their systematic yet flexible approach in capturing complex real-world behaviour, providing optimal forecasting performance, and explaining the intertwined dynamics of time series data [43]. VAR models flexibly reveal information about multivariate time series data and comprehensively explain endogenous variables through past values. Consequently, the study relies on secondary data (see S2 Appendix) acquired from SLBFE reports, CBSL reports, LFSDCS reports, and UGC-provided statistical data spanning from 1986 to 2022. The secondary data sources used for the study are presented in Table 2.

**Table 2. Data sources and definition of variables.**

| Variable | Definition | Measurement unit | Source |
|---|---|---|---|
| Migration | Migration | % Total number of male and female migration | Sri Lanka Bureau of Foreign Employment Statistics: http://www.slbfe.lk/page.php?LID=1&MID=275 |
| GDPPCI | GDP per capita income | Ln_GDPPCI | Central Bank Annual Reports: https://www.cbsl.gov.lk/en/publications/economic-and-financial-reports/annual-reports/annual-report-2022 |
| EG | Economic growth | % GDP annual growth rate | Central Bank Annual Reports: https://www.cbsl.gov.lk/en/publications/economic-and-financial-reports/annual-reports/annual-report-2022 |
| HE | Higher education | % Total graduates output | UGC Statistics Sri Lanka: https://www.ugc.ac.lk/index.php?option=com_content&view=article&id=2490%3Asri-lanka-university-statistics-2022&catid=55%3Areports&Itemid=42&lang=en |
| Unemployment | Unemployment | % Total numbers of male and female unemployment rate | Labour Force Survey Annual Reports: http://www.statistics.gov.lk/LabourForce/StaticalInformation/AnnualReports |

Source: Authors' compilations.

## Equation and VAR model

Each variable in a VAR model has a linear function of its prior values and the values of other variables [44, 45]. The mathematical Eq 1 of the VAR model, initially supported by [46, 47] represents the relationship of variables to elucidate macroeconomic and socioeconomic factors impacting migration in Sri Lanka.

$$Migration = a_1 + \sum_{j=1}^{j=p} b_{11}\, \text{GDPPCI}_{t-1} + \sum_{j=1}^{j=p} b_{12}\, \text{Unemployment}_{t-1} + \sum_{j=1}^{j=p} b_{13}\, \text{HE}_{t-1}$$
$$+ \sum_{j=1}^{j=p} b_{14}\, \text{EG}_{t-1} + u_{1t} \tag{1}$$

Where migration; GDPPCI, GDP per capita income (LKR); unemployment (rate); HE, higher education (total graduate output); EG, economic growth (real GDP growth rate); $a_1$, constant term; $b_{11}$, $b_{12}$, $b_{13}$ and $b_{14}$ coefficients; $u_{1t}$, error term; t-1, time; j = p, lag value.

The natural log of the first difference in migration; the natural log of the first difference in GDPPCI; the natural log of the first difference in unemployment; the natural log of the first difference in EG; VAR's output organises data by equation, with each "equation" corresponding to a dependent variable. Equations were developed for migration, GDPPCI, unemployment, HE and EG. The covariance matrix, denoted as (sigma), encapsulated the residuals resulting from VAR.

## Time series analysis

Plots of the time series of variables are illustrated in Fig 2. The values of the (i) graph represent HE, exhibiting irregular ups and downs behaviour. The GDPPCI (a) displays an upward-rising trend, while unemployment (c), EG (e), and the dependent variable migration (g) show downward falling trend. Graphs (b), (d), (f), and (h) depict time series changes characterised by irregular fluctuations. It's crucial to note that changes in the HE variable tend to fluctuate around a constant value. The phillips-perron test was performed to determine whether the variables were stationary or non-stationary, as demonstrated in S3 and S4 Appendices. According to S4 Appendix, the results indicated that migration, GDPPCI, unemployment, and EG are non-stationary, while S3 Appendix reveals that HE is stationary.

## Stationarity and stability assessment

The first difference in migration, GDPPCI, unemployment, and EG is depicted in S5 and S6 Appendices as migration, GDPPCI, Unemployment, and EG. Stationarity and stability are crucial assumptions for both individual data variables and the system, supporting the validity of multivariate and bivariate modelling within the VAR model framework. The stationarity of all variables has been established, as confirmed by the findings in Fig 3. The VAR model is deemed stable when all the eigenvalues of its coefficient matrix are below one, as per the formal definition [20, 46, 48]. The S7 and S8 Appendices revealed that the eigenvalues in the researchers' predictors fall within the unit circle; in Fig 3, VAR model stratifies the stability criterion.

## Model evaluation

The test for lag length detailed in S9 Appendix, recommended the use of four lags based on Akaik's information criterion (AIC), Schwarz's Bayesian information criterion (SBIC), and the Hannan and Quinn information criterion (HQIC). However, to ensure stability, researchers were advised to employ five lags, including the dependent variable. Jayawardhana, Anuththara [46] argued that the optimal lag length minimises the SBIC, AIC, and HQIC. When results were inconclusive, the decision was based on the minimum AIC criterion. One lag was

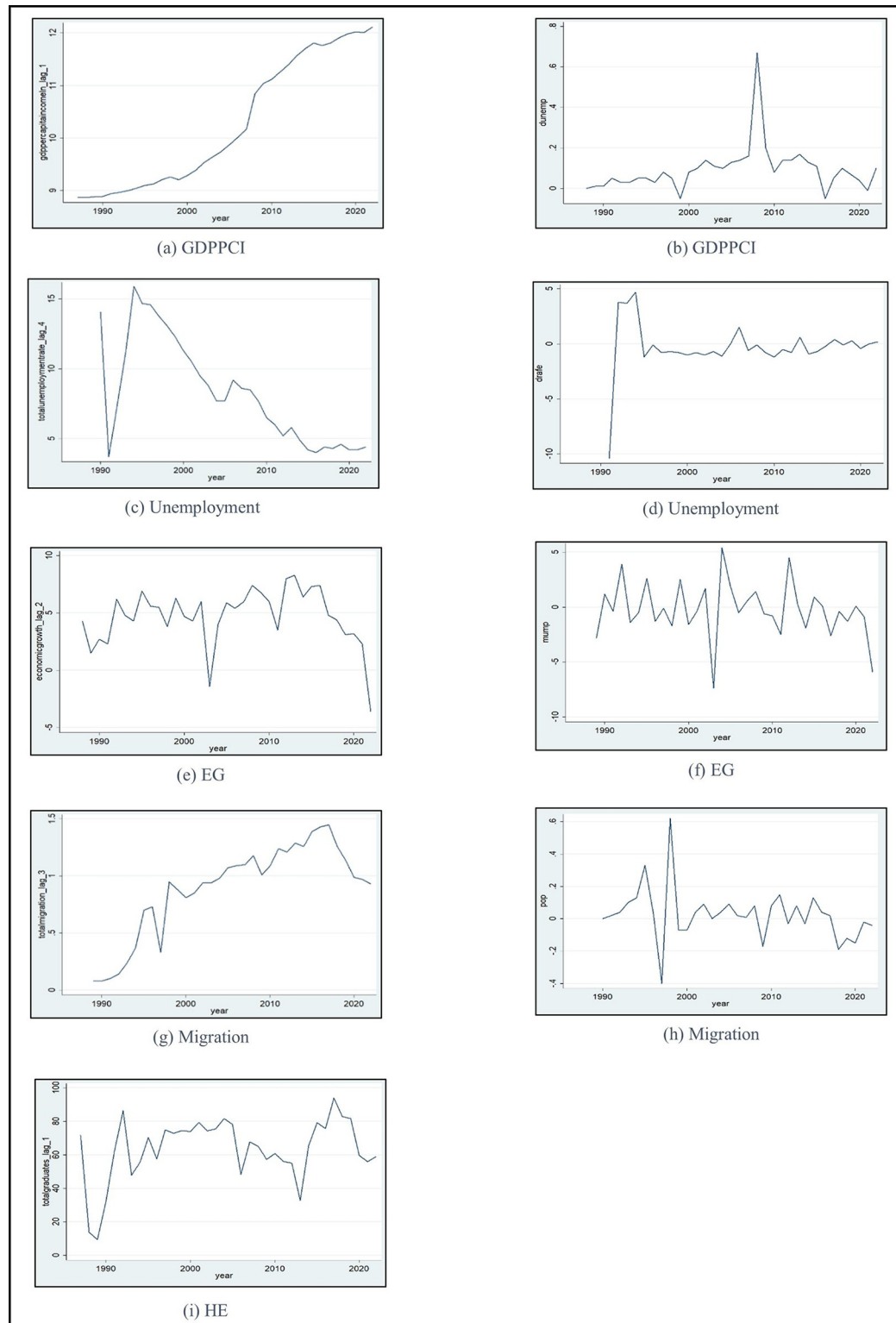

**Fig 2. Non-stationarity and stationarity time series.** Source: Authors' demonstration based on STATA.

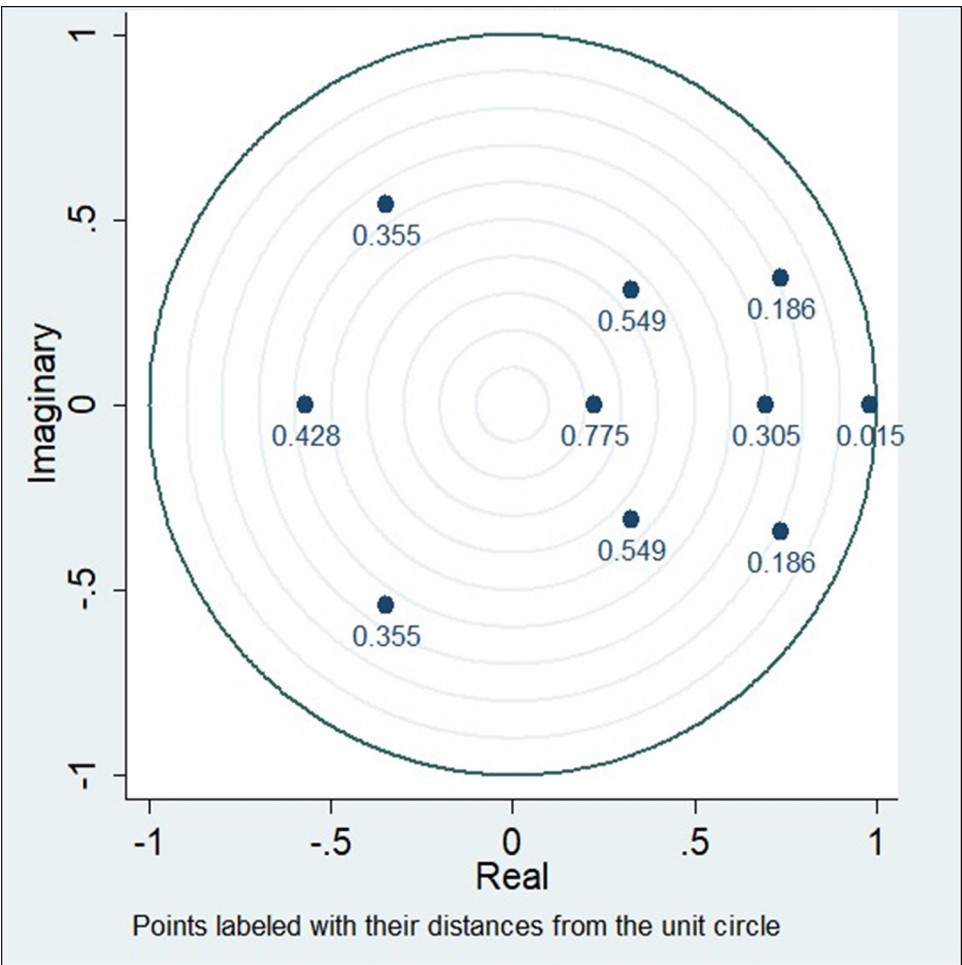

**Fig 3. Roots of the companion matrix.** Source: Authors' demonstration.

selected for GDPPCI and HE due to lower AIC, HQIC, and SBIC criteria. For EG, two lags were chosen, and four lags for unemployment, both minimizing AIC, HQIC, and SBIC. The dependent variable migration selected three lags, minimizing AIC and HQIC.

In addition, the S7 Appendix illustrates the identification of the model satisfying the stability condition with values less than one indicating the VAR model meets stability criteria. The data set was summarised in S7 Appendix displaying a mean of 0.0007464. S8 Appendix depicts the summarised error of the dataset, showing error term behaviour. Despite being compared with past and present error relations, no correlation between error and years was observed. The S7 Appendix presents the auto-correlation test, revealing no autocorrelation.

Khurshid [49] suggested the Granger causality test and cointegration analysis for determining causality among time series variables. A Johansen test examined cointegration, and the diagnostic test in S10 Appendix indicated that residuals showed no significant evidence of multicollinearity, autocorrelation, heteroscedasticity, model misspecification, or deviation from normality, with p-values above the 5% critical value. Eq 2, supported by [49, 50] represent the long-run relationship between variables in the cointegration relationship.

$$\Delta_{yt} = \alpha\beta'y_{t-1} + \sum_{1=I}^{m-1} \Pi_i y_{t-i} + \mu_t \tag{2}$$

In Eq 2, α signifies the long-run relationship in the VECM. The α matrix can calculated the cointegration connections. Π denotes a vector element (constant and trend), and μ$_t$ is a random error matrix.

## Results and discussion

This section unveils the key empirical findings of the study, followed by the VECM, VAR and Granger causality test outcomes. The objective was to assess the impact of GDPPCI, unemployment, HE, and EG on migration in Sri Lanka, thoroughly examining the autocorrelation of these variables and their influence on migration in Sri Lanka.

### Cointegration analysis

Cointegration test results in Table 3 rejected the hypothesis of no cointegration for unemployment, HE, and EG at the 1% significant level, and for GDPPCI at the 10% significant level. All t-values associated with the error-correlation term (ECT) from the conditional error-correlation model (ECM) were statistically significant [51], providing evidence of a long-run equilibrium relationship among the variables.

### Short-run and long-run analysis

Table 4 presents the results of both short-run and long-run analysis. The findings suggest that a decrease in year-on-year GDPPCI has driven the country towards elevated migration patterns, contributing to brain drain and labour market pressure. The long-term study emphasizes the substantial impact of Sri Lankan income utilization, revealing that a decrease in GDPPCI significantly increases migration decisions. Specifically, a 1% increase in GDPPCI has a long-term and significant migration effect, reducing it to 0.159 at the 10% significance level.

The analysis reveals a negative and significant long-term relationship between unemployment and migration, indicating that a 1% increase in the unemployment rate decreases migration by 5.062 at the 1% significance level. Similarly, a 1% rise in HE leads to a 9.232 increase in migration levels in the long run at a 10% significance level. Furthermore, empirical evidence confirms that EG negatively affects migration, showing a significant long-term relationship. A 1% increase in EG decreases migration by 0.562 at the 10% level. This underscores the importance of addressing these factors to mitigate brain drain and labour market pressure.

Following the long-run estimate analysis, a short-run estimate is evaluated to test migration effects. Table 4 outlines the short-run estimate outcomes, revealing crucial findings. GDPPCI, HE, and EG contribute to migration at a 10% significance level in the short run, while unemployment contributes to migration at a 1% significance level in the short run. These findings

**Table 3. Cointegration test results.**

| Predictors | Cointegration test | | |
|---|---|---|---|
| | EC Term | Std. Error | t-value |
| GDP per capita income | -0.022* | -0.031 | -7.480 |
| Total unemployment | 0.069*** | 0.010 | 6.790 |
| Total graduates | -0.009*** | 0.001 | -5.300 |
| Economic growth | -0.069*** | 0.013 | -5.080 |

Note: ***significant at the 1% level; **significant at the 5% level; *significant at the 10% level

Source: Authors' calculation based on STATA.

**Table 4. Results of VECM model.**

| Variable | Co-efficient | Std. Error | t-stat |
|---|---|---|---|
| **Long-run results** | | | |
| GDP per capita income | -0.159* | 0.089 | -1.79 |
| Total unemployment | -5.062*** | 0.552 | -9.17 |
| Total graduates | 9.232* | 10.586 | 0.87 |
| Economic growth | -0.562* | 1.918 | -0.29 |
| **Short-run results** | | | |
| GDP per capita income | -0.168* | 0.135 | -1.24 |
| Total unemployment | 2.940*** | 0.841 | 3.49 |
| Total graduates | -0.873* | 16.124 | -0.05 |
| Economic growth | -3.036* | 2.922 | -1.04 |
| **Ancillary parameters** | | | |
| Observations | 31 | | |
| Coint. eq. Chi-squared | 264.981 | | |
| Prob > chi$^2$ | 0.000 | | |
| Log-likelihood | -169.652 | | |

Note

***significant at the 1% level

**significant at the 5% level

*significant at the 10% level

Source: Authors' calculation based on STATA.

underscore the influence of low income levels, unemployment, a degraded education system, and slow economic progress on migration trends in Sri Lanka.

The results of the VAR model estimation results are presented in Table 5, where all variables are duly constrained. According to the VAR estimation, the lagged value of migration (0.3385437) significantly and positively influenced migration. The findings indicate that GDPPCI (-0.2993561) has a statistically significant negative impact on migration at the 1% level, while unemployment (2.280157) has a statistically significant positive impact at the 1% level. HE (10.03776) has a significant positive impact at the 10% level, and EG (-0.9723138) exhibits significant negative impact at the 10% level.

The study's reveals compelling evidence that unemployment and HE have a statistically significant positive impact on migration. In contrast, influence on migration, while GDPPCI and EG had a statistically significant negative impact on migration. These results align with Mohamed Aslam and Alibuhtto [36] findings on GDPPCI's negative impact on migration in Sri Lanka and Ayeni and Shaib [52] discovery of unemployment's negative impact on migration in Gambia. Additionally, the study supports Khan and Bin [40] observation of HE's positive impact on migration in China and Tipayalai [42] identification of EG's negative impact on migration in Thailand. The focus on time-series data within the Sri Lankan context strengthens the study's conclusions regarding the influential role of GDPPCI, unemployment, HE, and EG in migration decisions.

## The Granger causality Wald test

The Granger causality tests were conducted using time series data, and the results are presented in Table 6. Additionally, individual Granger causality tests for each variable were incorporated into Table 6. Following the approach outline by Jayawardhana, Jayathilaka [53], the

**Table 5. Estimation of the VAR model.**

| Variables | Coef. | SE | Z | P > (z) |
|---|---|---|---|---|
| Total migration | | | | |
| Total migration | | | | |
| L2 | 0.3385437** | 0.1614655 | 2.10 | 0.036 |
| GDP per capita income | | | | |
| Total migration | | | | |
| L2 | -0.2993561*** | 0.1109679 | -2.70 | 0.007 |
| Total unemployment | | | | |
| Total migration | | | | |
| L2 | 2.280157*** | 0.7300244 | 3.12 | 0.002 |
| Total graduates | | | | |
| Total migration | | | | |
| L2 | 10.03776* | 12.01794 | 0.84 | 0.404 |
| Economic growth | | | | |
| Total migration | | | | |
| L2 | -0.9723138* | 2.450189 | -0.40 | 0.691 |

Note

***significant at the 1% level

**significant at the 5% level

*significant at the 10% level

Source: Authors' calculation based on STATA.

VAR Granger causality test initially explores the individual causality of each VAR variable within each equation. Subsequently, it assesses the overall causality of all additional variables when considered collectively, providing a comprehensive evaluation of causal relationships within the study's framework.

Table 6 displays the Granger causality tests for the following equations, incorporating lagged variables such as Total migration, GDP per capita income, Total unemployment, Total graduates, and Economic growth. The Granger causality test is crucial tool for assessing the extent to which the lagged value of one variable contributes to the predicting another variable within the model [54]. Factors significantly contribute to predicting the targeted variable when the p-value associated with a factor's inclusion falls below the 0.05 threshold.

The causality Wald test results for GDP per capita income (0.0000), Total unemployment (0.0000), Total graduates (0.0017), and Economic growth (0.0407) support the predictors of GDPPCI, unemployment, HE, and EG, as the p-values were less than 0.05. These findings align with those of Islam and Khan [39] and Khan and Bin [40], indicating a Granger causal relationship between GDPPCI and HE and migration. This implies that low GDPPCI and an

**Table 6. Granger causality Wald tests.**

| Equation | Hypothesis | Excluded | Chi2 | Prob > Chi2 |
|---|---|---|---|---|
| GDP per capita income | H$_1$ | Total migration | 5429.064 | 0.0000 |
| Total unemployment | H$_2$ | Total migration | 1161.43 | 0.0000 |
| Total graduates | H$_3$ | Total migration | 28.11614 | 0.0017 |
| Economic growth | H$_4$ | Total migration | 18.96788 | 0.0407 |

Source: Authors' calculation based on STATA.

increasing demand for HE can significantly impact migration a lower GDPPCI and enhanced HE levels can exert an influential effect on migration in a country.

In this study, the four hypotheses, namely H1 ($0.0000 < 0.05$), H2 ($0.0000 < 0.05$), H3 ($0.0017 < 0.05$), and H4 ($0.0407 < 0.05$), each provide statistically significant evidence, affirming their validity and relevance within the research framework. In contrast to the findings of Boubtane, Coulibaly [55] and Qutb [56], the study identifies a substantial Granger causal relationship between unemployment and EG in relations to migration. The disparities in findings may arise from the present study's use of the latest data available for Sri Lanka, contrasting with past studies that analysed unemployment data on panel data from 1980 to 2005 across 22 OECD countries. Additionally, while earlier study examined EG in Egypt from 1980 to 2017, the present study focuses on the latest data spanning 1986 to 2022 in Sri Lanka. These variations in data and country contexts underscore the potential for divergent results.

Therefore, researchers in the present study can reject the null hypotheses for H1, H2, H3, and H4, as the probability associated with each hypotheses is less than the 0.05 critical value, indicating the presence of statistically significant findings in favour of the alternative hypotheses. These findings elucidate that GDPPCI, unemployment, HE, and EG influence migration, as the probability is less than the critical value (0.05).

The study's findings suggest that GDPPCI significantly influences migration in a country, as the probability associated with this impact is lower than the critical value. Therefore, researchers can reject the null hypothesis in this study that "GDPPCI does not cause migration", thus confirming a causal relationship between these two variables. As noted previously, by Islam and Khan [39], the findings of the Granger causality tests conducted on annual time series data emphasises the impact of GDPPCI on migration in the United States. The findings provide a compelling basis for rejecting the null hypothesis. The result strongly suggests a short-term causal relationship between GDPPCI and migration, but not vice versa. Hence, the conclusions drawn from the US data from developed countries offer insights that may apply to data from other developing countries, thereby advancing knowledge of migration in a different economic context.

A higher GDPPCI implies increased access to opportunities and higher incomes for individuals in the economy, leading to decreased migration in the country. These findings align with the analysis conducted by Ahmad and Arjumand [38], which was found to be significantly associated with migration in developing countries. Additionally, Uprety [57] study shows that when low-and middle-income groups face financial burdens in their lives, they tend to choose migration as a decision. This observation supports the argument that decreasing GDPPCI tends to push people to leave the country. For example, low-income countries such as Bangladesh, India, Indonesia, Kenya, Nepal, Pakistan, Sri Lanka and Vietnam represent a high emigration rate to find job opportunities or for best standards of living [58]. Currently, people's well-being in Sri Lanka has plummeted as tax rates and essential goods prices have risen. Therefore, the present study proves that income disparities can lead individuals to seek opportunities abroad. As a result, the study proves that the majority of people have left the country, creating labour market pressure.

Furthermore, the present study contradicts the prior study's findings, demonstrating that unemployment significantly influences migration. This necessitates the rejection of the null hypothesis, which asserted that "unemployment does not cause migration." It validates a causal link between these two variables conversely, Boubtane, Coulibaly [55] stated that the outcomes of the Granger causality test conducted on panel data across 22 OECD countries highlighted that unemployment significantly affected migration in none of the countries under study. Based on the findings, the study does not reject the null hypothesis, thereby affirming the absence of a causal relationship between these two variables. Consequently, the present study holds considerable significance, particularly in migration within developing countries.

In the context of China, studies on the patterns of migration provide added value to the literature in understanding the impacts of unemployment on the decision-making of migrants. As a developing country, China has not experienced increased poverty and unemployment in past decades. However, the majority of employment statuses will not be permanent [59]. Most of the jobs are temporary in China. Therefore, developed countries attract unemployed people or job-seeking individuals in China by providing job opportunities and other benefits. For example, Zhou, Roscigno [60] stated that the growing global demand for nurses has piqued the interest of developed countries in recruiting nurses from China. Consequently, a large number of employed and unemployed individuals apply for overseas nursing positions. This trend may endanger human development in the home country. Because people's human capital does not serve their home country, and that talent will take them to another country. If native countries do not clearly identify their human talent and do not provide them with opportunities, they are making a migration decision. The current situation in Sri Lanka serves as a poignant example of this challenge, where the country is grappling with the repercussions of not adequately recognising and providing opportunities for its human talent.

Moreover, findings suggest that HE can significantly influence migration in a country, as the probability associated with this impact is lower than the critical value. Therefore, researchers can reject the null hypothesis in this study that "HE does not cause migration." Hence, it can be inferred that a causal connection exists between these two variables. The findings of the Granger causality test on panel data by Khan and Bin [40] the impact of higher education on migration in China. The findings offer a strong foundation for rejecting the null hypothesis. Thus, earlier studies confirm the causal relationship between these two variables, as supported by the present research. As a result, this study stands out as unique in its scope, as it exclusively investigates the phenomenon of student migration in Sri Lanka.

Furthermore, the migration trend underlines the intricate relationship between economic difficulties and the aspirations for foreign education. Also Weber and Van Mol [9] stated, that many students consider their overseas education as their "ticket-to-migration." The increase in HE in a country can boost student migration because of the for foreign qualifications, especially postgraduate education in foreign institutes. Research in Russia by Rozhenkova [12] highlights that the increasing offerings of study abroad programs, dual-degree initiatives, and international collaborations at higher educational institutes positively and significantly impact student migration. Indeed, the above findings were confirmed in analyses by Levatino [61], who found that non-English-speaking countries such as Germany, Denmark, Norway, and France offer some of the world's most competitive PhD stipend programs and were significantly associated with student migration. As a result, when considering developing countries like Sri Lanka, the present study provides evidence that HE can increase the number of student migrations in the country.

Students in Asian countries such as India and Sri Lanka, whose popular destinations are Australia, Hong-Kong, and Singapore, all work in highly specific categories based on education [62]. For instance, in 2016, 78,107 Sri Lankan students enrolled in Australian universities, and 30,671 Sri Lankan students graduated from Australian universities in the same year [25]. The education migration industry, comprising education agents, recruiters, and money lenders, is highly responsible for facilitating the study abroad programme and providing developing countries' students with the opportunity to stay after graduation. These migrants will be able to become permanent residents or full citizens, while this is often directly impacted by native country brain drain and labour market pressure.

Likewise, findings suggest that EG can significantly influence migration in a country, as the probability associated with this impact is lower than the critical value. Therefore, researchers can reject the null hypothesis in this study that "EG does not granger cause migration", thus

confirming a causal relationship between these two variables. Similarly, the study by Qutb [56] which delved into the results of the Granger causality test conducted on time series data, underscores the influence of EG on migration patterns within the context of Egypt. These results indicate that there is no significant relationship between EG and migration. According to the findings, the study does not reject the null hypothesis, thus indicating there is no causal relationship between these two variables in the previous study. However, as opposed to the earlier studies, the present study confirms that EG has a significant relationship with migration.

The increase in EG can increase individual opportunities while reducing migration. In light of the study by Tipayalai [42], EG negatively and significantly impacts highly skilled emigrants. Nonetheless, IPS [32] postulated that if countries are experiencing weakened EG, that could substantially influence the migration decisions of individuals and families to seek better opportunities in developed countries to improve their living standards. Consequently, the researchers argue that increasing EG can decrease migration. The advancement of EG in developed countries attracts international workers and students; for example Sinatti [63] highlighted that a majority of African people do not return to their home countries after migrating. As a result, these countries are still developing. The country's development not only needs a financial source but also skilled employees. For example, Panagiotakopoulos [64] highlighted that in India, there is a concerning ratio of 1 doctor for every 2083 people, which indicates a shortage of medical professionals relative to the population size. This scarcity leads to prolonged wait times and limited access to specialized healthcare services. However, the United States boasts a more favorable ratio of 1 doctor for every 500 people. This reflects higher availability and improved access to healthcare services, potentially leading to better healthcare outcomes for the population. Therefore, without skilled peoples and knowledge, a home country cannot develop, and this is another factor in brain drain and labour market pressure in developing countries.

Moreover, according to Jayawardhana, Anuththara [46], pivotal outcomes were demonstrated through the Granger causality technique for examining the correlations between variables in both short and long-term contexts. The dependent variable is used to identify the calculation in the VAR result [65]. The VAR Granger tests were carried out to evaluate Granger causality by initially treating each variable as an independent one, calculated separately within each equation. This statistical test is a widely adopted approach for determining whether the lagged values of one variable can effectively forecast or predict the future values of other variables. Several VAR analyses show that a discernible causal relationship exists among GDPPCI, HE, and migration.

In contrast, some VAR analyses show no significant evident causal relationship among unemployment, EG, and migration. Therefore, it was demonstrated that total migration Granger influences GDPPCI, unemployment, HE, and EG. As a result, the VAR analysis conducted in this study furnishes compelling evidence, serving as a variable foundation for future research and affirming a substantial and significant causal relationship among GDPPCI, unemployment, HE, EG, and migration.

These challenges have yet to be taken up as a series of issues. Hence, the present country faces challenges in the domestic labour market and a brain drain. Historical evidence underscores the repercussions of ongoing issues. As per Saleem and Azad [66] the Philippines' nurse shortage led to the closure of hospital wards, restricting access to vital healthcare services and the departures of university lecturers from Mexico have compromised higher education quality impeding skilled workforce development and the country's global economic competitiveness. Thus, appropriate action should be taken by the responsible parties to avoid brain drain and protect the domestic labour market. According to the research conducted Bhardwaj and Sharma [4], apart from wage disparities and employment, other human-side factors also

influence skilled migration in the modern era. These include HE, employment opportunities, personal development, technological innovation, the standard of living, and the quality of work life [41]. Thus, researchers provide input for policy formulation and understand migration patterns more maturely. As a result, researchers suggest the term "human capital flight." This trend denotes that the investment made in educating and training these individuals does not yield benefits for the country of origin, such as Sri Lanka.

The departure of domestic employees seeking employment abroad can exert significant pressure on the current domestic labour market. Concurrently, there has been a noticeable rise in the workload of existing employees, leading to growing dissatisfaction and an increasingly stressful work environment. These factors can potentially be detrimental to the country's economy if left unaddressed. Sri Lanka has historically thrived on a well-staffed labour force, particularly in the professional sectors, and any decline in human resources availability among professionals is a matter of concern. The aforementioned facts prove that the present study can be used in all developing countries to identify factors that contribute to migration.

## Conclusion and policy recommendations

The primary objective of this study is to employ theoretical and empirical analysis to investigate how GDPPCI, unemployment, HE, and EG influence migration decisions in Sri Lanka. This aims to address the critical factors of brain drain and pressure in the domestic labour market. The secondary data was gathered annually through the SLBFE reports, CBSL reports, LFSDCS reports, and UGC-provided statistic data in Sri Lanka from 1986–2022 to accomplish the abovementioned objectives. The VECM model, VAR model and Granger Causality analysis were used to recognise the long-term relationship, impact and obtain a reliable picture of GDPPCI, unemployment, HE, and EG on migration.

The time-series estimations validate that low GDPPCI and weak EG predominantly influence migration in Sri Lanka. Based on the present study findings GDPPCI has played a pivotal role in rising migration, especially in the current economic crisis when people face financial hardship. For example, when citizen poverty and income inequality rise, individuals migrate to safeguard their families and alleviate hunger. Consequently, this tendency puts pressure on the domestic labour market as skilled and experienced professionals migrate, leaving organizations to deal with a manpower deficit. The subsequent burden placed on the remaining workforce increases stress, and the ensuing workload contributes to job dissatisfaction.

On the other hand, findings provide evidence for the presence of weak EG, Sri Lanka may witness business closures, and the combined impact of tax hikes and rising prices for essential commodities can serve as catalysts for emigration. Access to current data is crucial for a comprehensive grasp of the migration challenges, particularly given the labour market pressure stemming from workforce shortages and the country's overall economy. The increasing migration trend is creating a risk for overall development. Significant economic risks characterise Sri Lanka's current situation, which has led to a large population emigration. Unlike in the past, Sri Lanka is currently experiencing a drastically worse economy, which is causing important professionals like doctors and university lecturers to leave the country. This migration creates a serious threat to the country's healthcare and education system.

Furthermore, findings reported that increasing unemployment significantly influences migration, which can also impact brain drain and labour market pressure. The inability of individuals to find employment aligned with their skills and qualifications influences their desire to seek job opportunities abroad. Presently, a significant proportion of highly skilled employees from various organisations have migrated, creating a challenge to fill with adequately qualified candidates. The reason for this is that individuals, having endured prolonged

unemployment domestically, are increasingly inclined to migrate. When individuals feel that their native country does not sufficiently recognise and value their skill and knowledge, this trend is increased.

The findings suggest a positive influence of HE on migration, with a notably pronounced effect on the entry-level labour market in destination countries. The country's foreign education institutions offer foreign education facilities, which may influence this trend, and scholarship programs in developed countries serve to attract students because students from low-income countries like Sri Lanka face challenges in accessing university resources, scholarships, training, and post-education labour market opportunities, which can hinder their ability to enhance and utilise their knowledge effectively. This trend directly impacts the Sri Lankan brain drain. Following the provision of free primary education, a noticeable trend where students opt to pursue their highest studies abroad, eliminating any return on investment in their home country. However, beyond the stated capacity limitations, there is a deeper motivation that stems from the desire to migrate in order to pursue better opportunities for their well-being through migration. As a result, after migrating most of these students obtain citizenship or permanent residency in their destination country.

The study's findings and conclusions highlights the need for policymakers, bureaucrats, and the government in Sri Lanka to reevaluate the country's social welfare and development programs. The results show that push factors cause the majority of migration; therefore, policymakers should concentrate on resolving these issues such as lack of job opportunities, low wages and salaries, poor economic conditions, debt burden on the family, and lack of educational facilities.

Improving agricultural practices and increasing agricultural output ought to be the main priorities. Sri Lanka boasts fertile soil with a rich historical legacy of producing high quality agricultural products earning the well-deserved name of the "Eastern Granary" despite this, the CBSL reported the agricultural sector contributes in 2022, 8.75% of the country's GDP. Improving irrigation systems, access to credit for small-scale producers, and training programs on modern agricultural practices can eliminate poverty and income inequality in the country. By increasing agricultural productivity, employment and income-generating opportunities can be generated within the state, thereby reducing migration.

Second, funds should be allocated for the development of infrastructure, especially that related to telecommunications, electricity, and transportation. This will draw businesses and industries to Sri Lanka, boosting employment and the country's economy.

Third, implementing a strong local employment platform is a crucial step towards encouraging the production of some commodities domestically instead of relying on imports. One excellent example would be the effort to develop the "Valaichchenai paper Mill" through domestic paper production. This strategy not only makes a substantial contribution to the country's overall economic development but also to offering a wealth employment opportunity.

Finally, allocating funds to enhance educational facilities, such as resources for universities, scholarships, training programs, and post-education labour market opportunities, is essential. Also, raising the standard and applicability of HE to match the demands of Sri Lanka's workforce would take a coordinated effort. This closes the skills gap and produces a workforce more suited to meet the demands by ensuring a smoother entry for graduates into the workforce.

## Limitations

The study has certain limitations and shortcomings. First, research on brain drain and labour market pressure in Sri Lanka based on these variables is limited. Consequently, the present

study's literature has been developed by drawing upon existing studies from other countries and the Institute of policy studies (IPS) of Sri Lanka. Second, due to the unavailability of data predating 1986 because of the ethnic conflict and JVP terrorist attacks, coupled with the absence of a systematic data recording system, this study exclusively focuses on the period from 1986 to 2022. Third, this study solely utilised secondary data and, as a result, did not assess the migration aspirations of emigrants.

## Further recommendations

The findings establish a framework for future research in this field, offering a clear direction for further exploration. Future research may address these limitations by incorporating more recent data, capturing the evolving impact of GDPPCI, unemployment, HE, and EG on migration. Nevertheless, this study serves as a roadmap for expanding future research endeavours, offering new concepts and methodologies to enhance Sri Lankan macroeconomic and socio-economics projections. Studying the impact of macroeconomic and socioeconomic factors on migration can provide valuable insights for policymakers. By addressing these areas, future research can contribute to a more comprehensive understanding of the complex effects of GDPPCI, unemployment, HE, and EG on migration informing more effective policy strategies to aligned with SDGs.

## Supporting information

**S1 Appendix. Literature summary.**
(DOCX)

**S2 Appendix. Data file.**
(XLSX)

**S3 Appendix. Stationary variable–Higher education.**
(DOCX)

**S4 Appendix. Non-stationary variables.**
(DOCX)

**S5 Appendix. Generated non-stationary variables into stationary variables.**
(DOCX)

**S6 Appendix. Generated first difference for Migration.**
(DOCX)

**S7 Appendix. Eigenvalue stability condition summaries error and autocorrelation test.**
(DOCX)

**S8 Appendix. Summarized error.**
(DOCX)

**S9 Appendix. VAR optimum lag selection criteria.**
(DOCX)

**S10 Appendix. Diagnostic test result.**
(DOCX)

## Author Contributions

**Conceptualization:** Sandunima Kaluarachchi, Ruwan Jayathilaka.

**Data curation:** Sandunima Kaluarachchi.

**Formal analysis:** Sandunima Kaluarachchi, Ruwan Jayathilaka.

**Investigation:** Sandunima Kaluarachchi.

**Methodology:** Sandunima Kaluarachchi, Ruwan Jayathilaka.

**Software:** Sandunima Kaluarachchi.

**Supervision:** Ruwan Jayathilaka.

**Validation:** Sandunima Kaluarachchi, Ruwan Jayathilaka.

**Visualization:** Sandunima Kaluarachchi.

**Writing – original draft:** Sandunima Kaluarachchi, Ruwan Jayathilaka.

**Writing – review & editing:** Ruwan Jayathilaka.

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
