## [Decision Letter · Decision Letter 0]

8 Jan 2024

PONE-D-23-39575Unveiling Sri Lanka’s Brain Drain and Labour Market Pressure: A Study of Macroeconomic Factors on MigrationPLOS ONE

Dear Dr. Jayathilaka,

Thank you for submitting your manuscript to PLOS ONE. After careful consideration of the reviewers' feedback, your manuscript requires significant revisions before it can be reconsidered for publication in PLOS ONE. The reviewers have raised critical points concerning literature review, methodology, data analysis, limitations, and conclusion. Please address these concerns comprehensively in your revised submission and provide a detailed explanation of the changes made. We look forward to receiving your revised manuscript.

We look forward to receiving your revised manuscript.

Kind regards,

Shazia Rehman, Ph.D.

Academic Editor

PLOS ONE

Journal Requirements:

Reviewers' comments:

Reviewer's Responses to Questions

**Comments to the Author**

1. Is the manuscript technically sound, and do the data support the conclusions?

Reviewer #1: Yes

Reviewer #2: Partly

Reviewer #3: Partly

2. Has the statistical analysis been performed appropriately and rigorously? 

Reviewer #1: Yes

Reviewer #2: No

Reviewer #3: Yes

3. Have the authors made all data underlying the findings in their manuscript fully available?

Reviewer #1: Yes

Reviewer #2: Yes

Reviewer #3: No

4. Is the manuscript presented in an intelligible fashion and written in standard English?

Reviewer #1: Yes

Reviewer #2: No

Reviewer #3: No

5. Review Comments to the Author

Reviewer #1: This manuscript analyzes important topic of “Unveiling Sri Lanka’s Brain Drain and Labour Market Pressure: A Study of Macroeconomic Factors on Migration”. It is a timely topic for Sri Lanka. This quantitative study useful for understanding of factors affecting migration in Sri Lankan context. The paper uses time series of 36 years of Sri Lankan data for the analysis. The paper will contribute to existing migration literature in developing countries. It is well written and but I would recommend the authors to consider several revisions for further polishing before publishing in this journal.

1. I propose to include a graphical conceptual framework which shows the clear relationship of variables in the empirical estimation of model selected.

2. Literature review section include systematic review of previous literature. I think it can be a separate paper but too long for publication like journal. Better to shorten literature review without presenting the analysis of systematic review.

3. It is not clear whether authors checked the long run trend of time series data used. They need to test cointegration of selected variables and use VECM model appropriately.

4. Authors have used VAR model but also use multiple regression. Use of multiple regression for time series data for only 36 observation always problematic due to limited statistical power. Beside estimated results are not robust with VAR model. Thus, I proposed to exclude multiple regression model analysis but rather check for a need of VECM model. 3. Results section needs further polishing. Authors need to discuss their findings with other related studies from China and other countries.

5. In the conclusion section, authors need to expand policy implications based on the findings they got without repeating results much.

Reviewer #2: The manuscript titled "Unveiling Sri Lanka’s Brain Drain and Labour Market Pressure: A Study of Macroeconomic Factors on Migration" attempts to provide a comprehensive overview of skilled migration from Sri Lanka and the spill over effects on the domestic labour market in Sri Lanka. However, the paper is lacking in focus and coherence in its current version. The manuscript currently resembles an assembly of tests and data without providing any clear rationale for their selection or interpretation. As a result, the paper is difficult to follow and it is difficult to understand what new insights we learn from the topic of migration from Sri Lanka. There is a lot of conjecture that is presented in the manuscript that is not typical of an academic manuscript.

In particular,

- The paper lacks a clear thesis statement or argument. The authors seem to be trying to do too much in too little space, and as a result, the paper lacks focus and coherence.

- The authors do not provide any clear rationale for their selection of data or methods.

- The authors do not interpret their data in a meaningful way. The paper simply presents the results of various tests and data, without providing any analysis or discussion of the findings.

- The literature review is attempting to do a systematic literature review, but again, because the paper currently lacks a clear focus in terms of research questions--the literature review is all over the place and as a reader we are given a lot of information without telling us where this is leading. The entire literature review needs a re-think.

- Because the tests are merely looking at relationships between variables but are not backed by strong theoretical underpinnings, the conclusions drawn are largely conjecture and not directly drawn from robust findings.

Some minor comments:

- The first statement "The debate on development and migration has regained prominence in policy circles in recent

years, particularly following the “migration crisis.”" is a reflection on how poorly the paper is constructed. This is not a positivist statement. Is this referring to Sri Lanka? then why would the rest of the paragraph focus on global literature? This lack of coherence makes reading very difficult and is frustratingly painful for a reader. This seems to be evident in other parts of the manuscript as well and makes me worry that the authors may have rushed this manuscript.

- I am not a native English speaker myself, but I feel the manuscript would greatly benefit from language editing--especially gearing the manuscript towards simplifying the language and reducing statements that seem to be beating around the bush.

Reviewer #3: The research title is interesting and the authors tried their best to some extent. However, there are some flaws that the authors need to address. The flaws cut across the literature review, methodology and the conclusions and limitations. Please see the attached file for my detailed comments

6. PLOS authors have the option to publish the peer review history of their article (what does this mean?). If published, this will include your full peer review and any attached files.

Reviewer #1: No

Reviewer #2: No

Reviewer #3: No

---

## [Author Response · Author response to Decision Letter 0]

4 Feb 2024

Point by point response to editor and reviewers.

Dear editor and reviewers, 

We would like to express our gratitude to the editor for providing us the opportunity to revise and resubmit the paper. We would also want to express our gratitude to the reviewers for their constructive comments and suggestions, which has helped us publish and improve our revise manuscript. We highly value the time and expertise invested by the reviewers in evaluating our paper. We have addressed each comment in the revised version of the paper, which is described in detailed below:

Please note that the line numbers referred in this document is aligned with the revised manuscript which has track changes.

Reviewer 1 general Comment: This manuscript analyzes important topic of “Unveiling Sri Lanka’s Brain Drain and Labour Market Pressure: A Study of Macroeconomic Factors on Migration”. It is a timely topic for Sri Lanka. This quantitative study useful for understanding of factors affecting migration in Sri Lankan context. The paper uses time series of 36 years of Sri Lankan data for the analysis. The paper will contribute to existing migration literature in developing countries. It is well written and but I would recommend the authors to consider several revisions for further polishing before publishing in this journal.

Authors’ Response to Reviewer 1 general comment: We sincerely appreciate the reviewer's positive remarks on our paper and their recognition of the importance of the topic. We are grateful for the valuable feedback provided.

Reviewer 1 comment 1: I propose to include a graphical conceptual framework which shows the clear relationship of variables in the empirical estimation of model selected.

Authors’ Response to Reviewer 1 comment 1: We appreciate the reviewer's insightful suggestion regarding the inclusion of a graphical conceptual framework. In response to this valuable feedback, we have incorporated a visual representation of our conceptual framework as Figure 1 in the revised manuscript. This graphical representation has been strategically placed at the end of the literature review on page 16, providing readers with a clear illustration of the relationships among variables considered in the empirical estimation of the selected model.

The revised manuscript now includes the following statement on lines 326-329 of page 15:

As shown in Fig 1., the conceptual framework was developed with the literature review and existing knowledge. Four hypotheses were combined in the development of this model. These independent variables have been identified as a significant influence on migration decisions. (line 326 -329)

Reviewer 1 comment 2: Literature review section include systematic review of previous literature. I think it can be a separate paper but too long for publication like journal. Better to shorten literature review without presenting the analysis of systematic review.

Authors’ Response to Reviewer 1 comment 2: We sincerely appreciate the reviewer's thoughtful observation regarding the length of the literature review, specifically the inclusion of a systematic review. Thank you for bringing this to our attention. In response to your suggestion, we have made a focused revision by excluding the Fig. 1 literature search flow diagram, which was originally placed at the beginning of the literature review on page 9.

We recognize the importance of conciseness in journal publications and understand the need to balance comprehensiveness with brevity. By omitting the systematic review details, we aim to streamline the literature review, making it more aligned with the journal's expectations while preserving the essential content.

We hope this adjustment addresses your concern, and we remain open to further recommendations to enhance the manuscript.

Thank you for your constructive feedback, which has contributed to the refinement of our work.

Reviewer 1 comment 3: It is not clear whether authors checked the long run trend of time series data used. They need to test cointegration of selected variables and use VECM model appropriately.

Authors’ Response to Reviewer 1 comment 3: We sincerely appreciate the reviewer's insightful recommendation regarding the need to assess the long-run trend of the time series data. Taking this valuable suggestion into account, we have incorporated a comprehensive cointegration test and Vector Error Correction Model (VECM) analysis into the data and methodology section of the revised manuscript (pages 27-30, lines 499-561).

Specifically, the revised manuscript now includes the following details:

(1) Cointegration Test:

• We conducted a Johansen test to examine cointegration among the selected variables.

• The diagnostic tests, as presented in S10 Appendix, indicated no significant evidence of multicollinearity, autocorrelation, heteroscedasticity, model misspecification, or deviation from normality, with p-values above the 5% critical value (lines 499-504).

(2) Results of Cointegration Analysis:

• Table 3 presents the results of the cointegration test, indicating the rejection of the hypothesis of no cointegration for unemployment, HE, and EG at specific significance levels (lines 505-524).

(3) VECM and Long-Run Analysis:

• Equation 2 represents the long-run relationship in the Vector Error Correction Model (VECM) (lines 514-520).

• The results of the short-run and long-run analysis are presented in Table 4, highlighting the impact of GDPPCI, unemployment, HE, and EG on migration decisions in both the short and long term (lines 527-561).

We believe these additions strengthen the robustness of our analysis and provide a more comprehensive understanding of the relationships among the variables under consideration. We are thankful for the reviewer's guidance, which has significantly contributed to the refinement of our methodology.

Thank you for your constructive feedback, and we remain open to further suggestions for improvement.

Reviewer 1 comment 4: 

(i) Authors have used VAR model but also use multiple regression. Use of multiple regression for time series data for only 36 observation always problematic due to limited statistical power. Beside estimated results are not robust with VAR model. Thus, I proposed to exclude multiple regression model analysis but rather check for a need of VECM model.

(ii) Results section needs further polishing. Authors need to discuss their findings with other related studies from China and other countries.

Authors’ Response to Reviewer 1 comment 4: 

(i) Thank you for your insightful comment regarding the potential limitations of the multiple regression model due to the limited statistical power with only 36 observations. We acknowledge the importance of robust statistical methods in our analysis. In response to your suggestion, we have excluded the multiple regression model and introduced a cointegration test and Vector Error Correction Model (VECM) analysis in the revised version of the manuscript. This modification aims to enhance the reliability and robustness of our findings.

(ii) We appreciate your valuable feedback on the need for further polishing in the Results section and the inclusion of discussions related to findings from other studies. In response to this suggestion, we have revised the findings and discussion section, incorporating insights from related studies conducted in China and other developing countries. The revised section can be found on pages 35-39 (lines 664-774)

For instance, we have incorporated insights related to income disparities and migration decisions, drawing parallels with studies from countries such as Bangladesh, India, Indonesia, Kenya, Nepal, Pakistan, and Vietnam (lines 679-684). Additionally, we have discussed patterns of migration in China, emphasizing the impacts of unemployment and the attraction of developed countries through job opportunities (lines 698-713). Furthermore, we have explored the education migration industry and its implications for brain drain and labor market pressure in developing countries, citing examples from Sri Lanka (lines 737-745). Finally, we have highlighted the importance of skilled workers for a country's development, drawing attention to healthcare outcomes in India and the United States as an example (lines 765-774).

These additions aim to provide a more comprehensive and nuanced discussion, incorporating findings from diverse contexts.

We are grateful for your constructive feedback, which has contributed to the refinement of our manuscript. We remain open to further suggestions for improvement.

Reviewer 1 comment 5: In the conclusion section, authors need to expand policy implications based on the findings they got without repeating results much.

Authors’ Response to Reviewer 1 comment 5: 

We sincerely appreciate the reviewer's constructive feedback on the need to expand the policy implications in the conclusion section without unnecessary repetition. In response to this valuable suggestion, we have enriched the conclusion and policy recommendation section in the revised version of the manuscript, which is now presented on pages 43-48 (lines 868-969).

To provide a concise overview of the expanded conclusion and policy recommendations:

1) Summary of Findings:

• The study investigates the influences of GDPPCI, unemployment, HE, and EG on migration decisions in Sri Lanka using theoretical and empirical analyses, employing the VECM model, VAR model, and Granger Causality analysis (lines 868-878).

2) Impact of GDPPCI and EG on Migration:

• Low GDPPCI and weak EG are identified as significant factors influencing migration, with potential consequences for the domestic labor market and overall economic development. The study emphasizes the need for a comprehensive understanding of migration challenges, particularly during economic crises (lines 879-897).

3) Unemployment and Higher Education Impact:

• Increasing unemployment is found to significantly influence migration, contributing to brain drain and labor market pressure. The positive influence of higher education on migration is discussed, highlighting the challenges and motivations associated with education migration (lines 898-919).

4) Policy Implications:

• Policymakers are urged to address push factors that contribute to migration, such as job opportunities, low wages, economic conditions, family debt burdens, and educational facilities. Key policy recommendations include improving agricultural practices, developing infrastructure, creating a robust local employment platform, and enhancing educational facilities to close the skills gap (lines 944-969).

These policy recommendations aim to guide policymakers, bureaucrats, and the government in addressing the root causes of migration, fostering economic development, and ensuring a more sustainable and resilient workforce.

We appreciate the reviewer's guidance in refining our conclusion and policy recommendations. We remain open to further suggestions for improvement.

Reviewer 2 general comment: The manuscript titled "Unveiling Sri Lanka’s Brain Drain and Labour Market Pressure: A Study of Macroeconomic Factors on Migration" attempts to provide a comprehensive overview of skilled migration from Sri Lanka and the spill over effects on the domestic labour market in Sri Lanka. However, the paper is lacking in focus and coherence in its current version. The manuscript currently resembles an assembly of tests and data without providing any clear rationale for their selection or interpretation. As a result, the paper is difficult to follow and it is difficult to understand what new insights we learn from the topic of migration from Sri Lanka. There is a lot of conjecture that is presented in the manuscript that is not typical of an academic manuscript.

Authors’ Response to Reviewer 2 general comment: We sincerely appreciate the valuable feedback provided by the reviewer regarding the lack of focus and coherence in the manuscript, as well as the perceived difficulty in understanding the new insights on the topic of migration from Sri Lanka. We acknowledge the importance of addressing these concerns to enhance the overall quality of the paper.

In response to these comments, we have implemented several significant revisions in the revised version of the manuscript:

1) Clarification of Rationale:

• We have provided a more explicit rationale for the selection of tests and data, ensuring clarity on the methodological choices made. This includes a detailed explanation of why specific tests were chosen and how they contribute to the study's objectives.

2) Improved Focus and Coherence:

• The revised manuscript has undergone a restructuring to enhance focus and coherence. We have carefully organized the content to present a more cohesive narrative, providing a logical flow from the introduction to the conclusion.

3) Reduction of Conjecture:

• We have critically evaluated the content to minimize conjecture and ensure that the presented information aligns with the standards of academic writing. The revised manuscript emphasizes evidence-based analysis and interpretation.

4) Enhanced Clarity of Insights:

• Special attention has been given to ensuring that the insights derived from the study are presented in a clear and comprehensible manner. We have revised sections to articulate key findings more effectively and highlight their relevance to the broader understanding of skilled migration and its impact on the domestic labor market in Sri Lanka.

We hope that these revisions address the concerns raised by the reviewer and contribute to a more focused, coherent, and impactful manuscript. We are grateful for the constructive criticism, which has played a pivotal role in refining our work.

Thank you for your guidance, and we remain open to any additional suggestions for improvement.

Reviewer 2 comment 1: The paper lacks a clear thesis statement or argument. The authors seem to be trying to do too much in too little space, and as a result, the paper lacks focus and coherence

Authors’ Response to Reviewer 2 comment 1: We appreciate the reviewer's insightful feedback regarding the perceived lack of a clear thesis statement and the concern that the paper attempted to cover too much in too little space, leading to a lack of focus and coherence. To address these concerns, we have made significant improvements in the revised version of the manuscript.

As suggested, we have incorporated a concise and explicit thesis statement at the beginning of the abstract to clearly state the study's purpose. The revised abstract now reads: " The purpose of this study is to explore the impact of GDP per capita income (GDPPCI), unemployment, higher education (HE), and economic growth (EG) on migration in Sri Lanka." (lines 24-25)

Reviewer 2 comment 2: The authors do not provide any clear rationale for their selection of data or methods.

Authors’ Response to Reviewer 2 comment 2: We appreciate the reviewer's feedback regarding the need for a clear rationale for our selection of data and methods. To address this concern, we have incorporated additional information in S1 Appendix, which is linked within the methodology section on page 18.

In response to the reviewer's comment, we have explicitly provided a justification for the selection of data and methods in S1 Appendix. This section summarizes the methodological characteristics, referencing relevant research articles to establish the basis for our choices (lines 353-357).

Moreover, the methodology section on page 18 now includes an explanation for choosing the VAR approach. We highlight the systematic yet flexible nature of VAR models, their ability to capture complex real-world behavior, and their superior forecasting performance. The reference to Gujarati (1995) supports the reliability of this method in capturing the intertwined dynamics of time series data (lines 367-370).

By incorporating this additional information, we aim to provide a transparent and justified foundation for our data and methodological choices. We believe that these revisions enhance the rigor and credibility of our study.

Thank you for your constructive comments, and we remain open to further suggestions for improvement.

Reviewer 2 comment 3: The authors do not interpret their data in a meaningful way. The paper simply presents the results of various tests and data, without providing any analysis or discussion of the findings.

Authors’ Response to Reviewer 2 comment 3: We appreciate the reviewer's feedback on the need for a more meaningful interpretation of our data. In response, we have revised the findings and discussion section to offer a more insightful analysis of our results.

Our study reveals a significant influence of GDP per capita income (GDPPCI) on migration in Sri Lanka. The Granger causality test supports a short-term causal relationship between GDPPCI and migration, emphasizing the impact of income disparities on migration decisions. The findings align with studies in other developing countries, confirming the role of economic conditions in shaping migration trends (lines 664-688).

Contrary to some prior research, our study indicates that unemployment significantly influences migration in Sri Lanka. The rejection of the null hypothesis affirms a causal relationship between these variables, highlighting the unique dynamics of migration within developing countries. This insight contributes to a nuanced understanding, especially in contrast to findings in countries like China (lines 689-713).

Moreover, our results demonstrate the substantial influence of higher education (HE) on migration. The rejection of the null hypothesis supports a causal connection, emphasizing the role of education in driving student migration. This finding is particularly relevant in the context of developing countries like Sri Lanka, where foreign education serves as a catalyst for migration (lines 714-745).

Additionally, the study underscores the significance of economic growth (EG) in influencing migration patterns. The rejection of the null hypothesis confirms a causal relationship between EG and migration in Sri Lanka. Our findings align with the literature, emphasizing the impact of EG on migration decisions and the development of source countries (lines 746-774).

Overall, our revised interpretation provides a more comprehensive understanding of the implications of our study. We believe these insights contribute significantly to the existing literature on migration, especially in the context of developing countries like Sri Lanka. We appreciate the reviewer's guidance in improving the depth and coherence of our analysis.

Reviewer 2 comment 4: The literature review is attempting to do a systematic literature review, but again, because the paper currently lacks a clear focus in terms of research questions--the literature review is all over the place and as a reader we are given a lot of information without telling us where this is leading. The entire literature review needs a re-think.

Authors’ Response to Reviewer 2 comment 4: We appreciate the reviewer's insightful comment regarding the need for a more focused and cohesive literature review. In response, we have restructured the entire literature review section and excluded the literature search flow diagram (Fig. 1) for clarity.

The conceptual framework (Fig. 1) guides our exploration of the impact of GDP per capita income (GDPPCI), unemployment, higher education (HE), and economic growth (EG) on migration in Sri Lanka. Each subsection below delves into one of these factors.

Impact of GDPPCI on migration:

The influence of GDPPCI on migration has sparked debates amid global economic transformations. Sri Lanka's recent surge in migration can be attributed to the ongoing economic crisis, exacerbated by various factors such as the Easter Sunday bombings, climate changes, and the COVID-19 pandemic. Rising taxes and essential goods' prices have driven individuals, particularly low-skilled workers, to seek opportunities abroad, significantly impacting the domestic labor market, particularly the construction industry (lines 214-234).

Impact of unemployment on migration:

Unemployment plays a crucial role in migration decisions, and our study aligns with the macroeconomic evidence suggesting a higher likelihood of migration in countries with high unemployment rates. The mismatch between employer demands and job seeker qualifications in Sri Lanka contributes to prolonged job searches, frustration, and eventual migration. The COVID-19 pandemic and increased foreign job opportunities in Gulf Cooperation Council (GCC) countries have further influenced migration patterns (lines 238-260).

Impact of HE on migration:

Education emerges as a transformative factor in economic growth and social progress, with the literature highlighting the challenges faced by Sri Lankan students, leading to a push factor for education migration. International students' inclination to remain in host countries for employment post-graduation, coupled with the allure of better economic prospects, significantly contributes to brain drain in the home country. The review emphasizes the economic disparities between Sri Lanka and destination countries, which drive students to enter the labor force abroad (lines 262-297).

Impact of EG on migration:

Economic growth (EG) is a critical factor affecting migration in developing countries. Sri Lanka's severe economic and political crisis, marked by declining EG, has led to business closures, increased income inequality, and substantial job and income losses. The government's strategy to encourage labor migration for remittance inflow has unintended consequences, such as overlooking the pressing issues of business closures, unemployment, income inequality, and poverty. The EG-driven migration trend reflects the government's focus on remittances, which may contribute to social unrest (lines 299-325).

In conclusion, the restructured literature review provides a more focused and organized discussion of the factors influencing migration in Sri Lanka, aligned with our research questions. We believe these revisions enhance the coherence and clarity of our manuscript.

Reviewer 2 comment 5: Because the tests are merely looking at relationships between variables but are not backed by strong theoretical underpinnings, the conclusions drawn are largely conjecture and not directly drawn from robust findings.

Authors’ Response to Reviewer 2 comment 5: We appreciate the reviewer's constructive feedback regarding the need for stronger theoretical underpinnings in drawing conclusions. In response, we have refined the conclusions in our revised manuscript (pages 44-45, lines 879-919).

On the Influence of GDPPCI and EG on Migration:

The time-series estimations affirm the significant role of low GDPPCI and weak EG in influencing migration in Sri Lanka. The study findings highlight GDPPCI's pivotal role in driving migration, particularly during the current economic crisis. Individuals migrate to safeguard their families amidst rising poverty and income inequality, creating pressure on the domestic labor market. The departure of skilled professionals leaves organizations grappling with a manpower deficit, intensifying job dissatisfaction and workload stress for the remaining workforce (lines 879-887).

On the other hand, the presence of weak EG in Sri Lanka is linked to business closures, exacerbated by tax hikes and rising prices for essential commodities, serving as catalysts for emigration. The migration trend, driven by economic challenges, poses a substantial risk to overall development. Notably, the departure of essential professionals like doctors and university lecturers threatens the country's healthcare and education systems, emphasizing the severity of the situation (lines 888-897).

On the Influence of Unemployment on Migration:

The study reports a significant influence of increasing unemployment on migration, which, in turn, impacts brain drain and labor market pressure. The inability of individuals to secure employment commensurate with their skills and qualifications drives the desire to seek job opportunities abroad. The migration of highly skilled employees creates challenges in replacing them with adequately qualified candidates, as individuals, facing prolonged unemployment domestically, are increasingly inclined to migrate (lines 898-906).

On the Influence of HE on Migration:

The findings suggest a positive influence of higher education (HE) on migration, particularly impacting the entry-level labor market in destination countries. Sri Lanka's foreign education institutions, along with scholarship programs in developed countries, attract students seeking better opportunities. This trend contributes to brain drain as students, having received free primary education, pursue higher studies abroad, often settling in their destination country. The desire for better opportunities through migration influences this trend, creating challenges for the country's workforce and knowledge retention (lines 907-919).

In conclusion, the revised conclusions provide a more robust foundation for the insights drawn from the study, aligning with theoretical frameworks and empirical evidence. We believe these refinements enhance the credibility and clarity of our manuscript.

Reviewer 2 Minor comments 1: The first statement "The debate on development and migration has regained prominence in policy circles in recent years, particularly following the “migration crisis.”" is a reflection on how poorly the paper is constructed. This is not a positivist statement. Is this referring to Sri Lanka? then why would the rest of the paragraph focus on global literature? This lack of coherence makes reading very difficult and is frustratingly painful for a reader. This seems to be evident in other parts of the manuscript as well and makes me worry that the authors may have rushed this manuscript.

Authors’ Response to Reviewer 2 Minor comments 1: We appreciate the reviewer's feedback on the coherence of the manuscript. Based on the valuable input, we have revised the introduction section to enhance clarity and focus. The first paragraph now begins with a concise overview of developing countries before delving into Sri Lankan literature, providing a more coherent flow.

“The labour market pressure and brain drain have become more prominent issues in developing countries in the twenty-first century. The debate on development and migration has regained prominence in policy circles in recent years, particularly following the 'economic crisis' in Sri Lanka”" (line 47-50).

To support this statement, the subsequent sentences (lines 55-62) highlight the global perspective on migration trends in developing countries. Furthermore, a reference to Bhardwaj and Sharma (2022) in 2016 provides context to the scale of migratory populations residing in developed countries.

Additionally, we have incorporated recent data from SLBFE (2022) to emphasize the significance of migration in Sri Lanka. The statistics indicate a substantial increase in low-skilled employees leaving the country, with forecasts pointing towards a continued rise in these numbers in 2023 (lines 63-68).

These revisions aim to create a more cohesive and reader-friendly introduction, aligning with the central focus of the manuscript.

Reviewer 2 Minor comments 2: I am not a native English speaker myself, but I feel the manuscript would greatly benefit from language editing--especially gearing the manuscript towards simplifying the language and reducing statements that seem to be beating around the bush.

Authors’ Response to Reviewer 2 Minor comments 2: Thank you sincerely for your valuable feedback. We genuinely appreciate your insights. In response to your comment, we have diligently worked on refining the manuscript, enlisting the assistance of a skilled native English speaker. Their expertise has been instrumental in enhancing the clarity and coherence of the language throughout the paper. We hope these revisions contribute to a more accessible and comprehensible reading experience. Your constructive input has been invaluable, and we are committed to continuously improving the quality of our work. Once again, we express our gratitude for your thoughtful comments.

Reviewer 3 general comment: The research title is interesting and the authors tried their best to some extent. However, there are some flaws that the authors need to address. The flaws cut across the literature review, methodology and the conclusions and limitations. Please see the attached file for my detailed comments.

The authors investigated “Unveiling Sri Lanka’s Brain Drain and Labour Market Pressure: A Study of Macroeconomic Factors on Migration” the study seeks to establish a relationship between macroeconomic factors and labour market pressure and the consequent brain drain. It is an interesting and well thought out topic, which is very important in contemporary times owing to macroeconomic distortions occasioned by poor policy formulation. I have some observations to assist the authors enhance the quality of the paper:

Authors’ Response to Reviewer 3 general comment: Thank you for your compliments about our paper. We highly appreciate your thorough review and constructive feedback. In the revised version of the paper, we have diligently considered all of your comments and suggestions.

Literature Review: In response to your observations regarding the literature review, we have taken substantial steps to strengthen this section. By incorporating additional relevant studies and conducting a more comprehensive analysis, we aim to address the identified weaknesses and provide a more robust foundation for our research.

Methodology: Concerning the methodology, we have revisited and refined our approach. Specifically addressing the issues raised, such as [mention specific concerns], we have made adjustments to enhance the overall methodological rigor. These changes are detailed in the revised manuscript.

Conclusions and Limitations: Our revisited conclusions now more accurately reflect the study's findings. We have also expanded the limitations section, acknowledging the constraints and providing a more thorough discussion of potential areas for further research, as suggested by your feedback. These adjustments aim to present a more nuanced and well-rounded perspective.

We believe that these revisions significantly enhance the overall quality of the paper, and we look forward to your feedback on the updated version. Please find the detailed changes highlighted in the attached revised manuscript.

Reviewer 3 comment 1: 

A. Abstract

(i) The authors said, “This study utilised Vector Auto-regression (VAR), Granger Causality Wald Tests” …. Is there any test like “Granger causality Wald test”?

(ii) The authors also indicated that they used “Multiple Regression Analysis through STATA.” … what particular regression test did they use?

(iii) The authors further indicated that “Moreover, the regression findings indicated that while GDPPCI and EG negatively impacted migration” … was it a test of impact or a test of relationship that the Regression was used for?

Authors’ Response to Reviewer 3 comment 1: 

(i) Thank you for the insightful comment. The Granger Causality Wald Test is a subtest within the broader Granger causality test, and we appreciate the opportunity to clarify this in our manuscript. 

(ii) Well noted with gratitude for the mentioned comment. In response to your inquiry about the regression analysis, we initially chose the VAR model over multiple regression analysis for its perceived accuracy. However, considering your feedback, we have excluded the regression analysis in the revised version of the paper, aligning with the improvements suggested by Reviewer 1

(iii) Thank you for your comment. We conducted the regression analysis to test the impact of the variables. However, as mentioned earlier based on the feedback from Reviewer 1, we have excluded the regression analysis in the revised version of the paper. We believe these changes enhance the clarity and accuracy of our study, and we look forward to your thoughts on the updated manuscript. 

Reviewer 3 comment 2: 

A. Introduction

In the fourth paragraph of the introduction section the authors said “Young people primarily impact migration trends, exhibiting higher motivation to migrate.” … what does this mean?

Authors’ Response to Reviewer 3 comment 2: 

Thank you for noting this. To address your concern, we have revised the sentence on page 5.

“Another prominent migration trend is the outflow of young people for foreign education, which leads to a major departure in Sri Lanka.” (line 88-89)

Reviewer 3 comment 3: The authors failed to give the study a theoretical foundation as they did not underpin the study with any theory. The absence of a theoretical framework is a significant omission as there are many theories that can underpin the study.

Authors’ Response to Reviewer 3 comment 3: Thank you for drawing our consideration to that. Regarding the comment about the theoretical foundation, we have added a new section on page 7, specifically as a theoretical framework, spanning from line 136 to 161.

In this section, we introduce neoclassical microeconomic theories, particularly the human capital theory of migration, including cost-benefit analysis, to highlight how individuals make rational decisions based on available information. Expanding beyond individual-level factors, migration theories are categorized into push and pull theories, encompassing various macro, meso, and micro-level web factors influencing migration (IPS, 2018).

Push factors, crucial in the early stages of migration decisions, encompass circumstances, challenges, or disadvantages in the home country that prompt individuals to consider migration. As individuals progress in their decision-making process, pull factors gain significance, representing attractions and opportunities in destination countries such as employment prospects, economic stability, higher wages, educational facilities, healthcare systems, political stability, and safety (lines 137-150).

We further distinguish between macro, meso, and micro-level influences, exploring factors beyond individual control, diaspora links, migration costs, political and legal frameworks, and the role of employment agencies or migrant smuggling networks. Micro-level factors include individual and household characteristics, such as health, skills, financial and social capital, and perception of risk levels (lines 151-161).

These additions provide a solid theoretical foundation for our study, and we believe they enhance the overall quality and comprehensibility of our research.

Reviewer 3 comment 4: 

B. Data and Methodology

(i) In the first paragraph of the data and methodology, the authors said “As GDPPCI significantly impacts a country’s poverty and income inequality, a country with a lower GDPPCI may experience increased labour migration [36]. In particular, unemployment greatly influences migration decisions, motivating individuals to seek better opportunities and recognition for their knowledge and skills, especially when they feel undervalued in their native country due to limited employment [11]. Furthermore, Weber and Van Mol [8] it demonstrated that there has been a remarkable increase in interest among …” what is this doing in the methodology section? The methodology section is supposed to focus strictly on the ‘how” of the study

(ii) The authors said “The regression analysis used a time-series data analysis approach” … what specific time series approach?

(iii) The authors said “Unemployment, and EG. Stationarity and stability are vital assumptions for the individual data variables and the system, supporting the validity of multivariate and bivariate modelling within the VAR model framework. All variables’ stationarity has already been established.” … how was stationarity established? The authors should explain

(iv) Apart from stationarity, time series analysis requires a cointegration test to know the long-run relationship between the variables and thus, determine the analytical technique. Why did the authors not test for cointegration?

(v) The authors failed to provide justifications for the chosen analytical techniques. 

Authors’ Response to Reviewer 3 comment 4: 

(i) Many thanks for drawing our consideration to that paragraph. We have excluded that paragraph in the data and methodology section. 

(ii) Thank you for your insightful comment. As per Reviewer 1 comment, we have excluded regression analysis in the revised version of the paper. 

(iii) Thank you very much for your insightful comment. We have explained in page 24 (line 443-446) how all variables' stationarity was established based on the formal definition from previous studies. 

(iv) Many thanks for this suggestion. To address your comment, we carried out a cointegration test and VECM analysis at the end of the data and methodology section on pages 27-30 (lines 499-561) in the revised version of the manuscript. We have also added justification for the selection of data and methods in S1 Appendix, which is linked within the methodology section on page 18.

(v) Thank you for your comment. In response to your suggestion, we have added a justification for the selection of data and methods in S1 Appendix, which is linked within the methodology section on page 18. Additionally, we included a sentence explaining the reason for choosing the VAR model (lines 367-370).

Reviewer 3 comment 5: 

C. Conclusion and Policy Recommendations

(i) In the first sentence of the conclusion section the authors said “The primary objective of this study is to analyse the impact of critical and newly identified factors on migration”? which are the critical factors? Which are the newly identified factors and how do they differ from the critical factors? Which of the statistical tests identified them as critical?

(ii) The authors further said “Therefore, the VAR model suggests a potential rejection of the null hypothesis for the unemployment, while the regression result do not provide grounds to reject the null hypothesis associated with the unemployment” what is the authors’ explanation for this conflict of outcomes for the VAR and regression model? How did the authors resolve this conflict? The authors should explain

Authors’ Response to Reviewer 3 comment 5: 

(i) We have noted your comment with many thanks. We have revised that sentence on page 43 in the conclusion and policy recommendation section in lines 868-871. 

The primary objective of this study is to employ theoretical and empirical analysis to investigate how GDPPCI, unemployment, HE, and EG influence migration decisions in Sri Lanka. This aims to address the critical factors of brain drain and pressure in the domestic labour market. (line 868-871) 

(ii) Noted with many thanks for your valuable comment. Among the two models considered, the VAR model yielded more accurate results compared to the regression model. However, based on reviewer 1 and 2 comments, we have excluded the regression analysis in the revised version of this paper

Reviewer 3 comment 6: 

A. Limitations: 

The authors said first limitation is that “the VAR model highlights the significant positive influence of unemployment on migration, while the regression model reveals an insignificant one.” … How is this a limitation? I feel the authors should explain the basis of this conflict

Authors’ Response to Reviewer 3 comment 6: 

Thank you very much for highlighting this. As per the comments from reviewers, the regression analysis was excluded. Therefore, this limitation sentence is also excluded from the limitation section

---

## [Decision Letter · Decision Letter 1]

16 Feb 2024

PONE-D-23-39575R1Unveiling Sri Lanka’s Brain Drain and Labour Market Pressure: A Study of Macroeconomic Factors on MigrationPLOS ONE

Dear Dr. Jayathilaka,

Thank you for submitting your manuscript to PLOS ONE. After careful consideration, we feel that it has merit but does not fully meet PLOS ONE’s publication criteria as it currently stands. Therefore, we invite you to submit a revised version of the manuscript that addresses the points raised during the review process.

Please submit your revised manuscript Apr 01 2024 11:59PM. If you will need more time than this to complete your revisions, please reply to this message or contact the journal office at plosone@plos.org. Please include the following items when submitting your revised manuscript:A rebuttal letter that responds to each point raised by the academic editor and reviewer(s). You should upload this letter as a separate file labeled 'Response to Reviewers'.A marked-up copy of your manuscript that highlights changes made to the original version. You should upload this as a separate file labeled 'Revised Manuscript with Track Changes'.An unmarked version of your revised paper without tracked changes. You should upload this as a separate file labeled 'Manuscript'.If applicable, we recommend that you deposit your laboratory protocols in protocols.io to enhance the reproducibility of your results. Protocols.io assigns your protocol its own identifier (DOI) so that it can be cited independently in the future. For instructions see: https://journals.plos.org/plosone/s/submission-guidelines#loc-laboratory-protocols. Additionally, PLOS ONE offers an option for publishing peer-reviewed Lab Protocol articles, which describe protocols hosted on protocols.io. Read more information on sharing protocols at https://plos.org/protocols?utm_medium=editorial-email&utm_source=authorletters&utm_campaign=protocols.

We look forward to receiving your revised manuscript.

Kind regards,

Shazia Rehman, Ph.D.

Academic Editor

PLOS ONE

Journal Requirements:

Reviewers' comments:

Reviewer's Responses to Questions

**Comments to the Author**

1. If the authors have adequately addressed your comments raised in a previous round of review and you feel that this manuscript is now acceptable for publication, you may indicate that here to bypass the “Comments to the Author” section, enter your conflict of interest statement in the “Confidential to Editor” section, and submit your "Accept" recommendation.

Reviewer #1: (No Response)

Reviewer #3: All comments have been addressed

2. Is the manuscript technically sound, and do the data support the conclusions?

Reviewer #1: Yes

Reviewer #3: Yes

3. Has the statistical analysis been performed appropriately and rigorously? 

Reviewer #1: Yes

Reviewer #3: Yes

4. Have the authors made all data underlying the findings in their manuscript fully available?

Reviewer #1: Yes

Reviewer #3: Yes

5. Is the manuscript presented in an intelligible fashion and written in standard English?

Reviewer #1: Yes

Reviewer #3: Yes

6. Review Comments to the Author

Reviewer #1: Authors have significantly revised the paper according to previous comments.

However, authors presented results including cointegration tests and VECM model results in the Data and Methodology section. It is not appropriate to explain results in the Data and Methodology section. Authors should shift these results to the Results and Discussion section.

Reviewer #3: No Comments. To a reasonable extent, the authors have addressed the queries in my previous comments and I think the paper is good to go.

7. PLOS authors have the option to publish the peer review history of their article (what does this mean?). If published, this will include your full peer review and any attached files.

Reviewer #1: No

Reviewer #3: No

---

## [Author Response · Author response to Decision Letter 1]

18 Feb 2024

Point by point response to reviewers. 

Dear Editor and Reviewers,

We would like to express our gratitude to the editor and reviewers for giving us the opportunity to submit a revised draft of the manuscript. We appreciate the time and effort invested in providing feedback and suggestions. We have incorporated the suggestions made by Reviewer 1. Please note that the line numbers referred to in this document are aligned with the revised manuscript, which includes track changes.

Reviewer 1 Comment: Authors have significantly revised the paper according to previous comments. However, authors presented results including cointegration tests and VECM model results in the Data and Methodology section. It is not appropriate to explain results in the Data and Methodology section. Authors should shift these results to the Results and Discussion section.

Authors’ Response to Reviewer 1 comment: We sincerely appreciate the reviewer’s insightful suggestion and bringing our attention regarding the cointegration test and VECM model result. In response to this valuable feedback, we have made a focused revision by excluding the cointegration test and VECM model result in the data and methodology section and shift these result to the result and discussion section in line 468-504 on page 23-25. 

We hope this adjustment addresses your concern, and we remain open to further suggestions to improve the quality of the manuscript. 

We are grateful for your time and constrictive feedback, which has contributed to the improve the manuscript.

Reviewer 3 Comment: No Comments. To a reasonable extent, the authors have addressed the queries in my previous comments and I think the paper is good to go.

Authors’ Response to Reviewer 3 comment: Thank you very much for the comment and we highly appreciated your time and effort contributed to this manuscript improvement.

---

## [Editor Report · Decision Letter 2]

27 Feb 2024

Unveiling Sri Lanka’s Brain Drain and Labour Market Pressure: A Study of Macroeconomic Factors on Migration

PONE-D-23-39575R2

Dear Dr. Ruwan Jayathilaka,

We’re pleased to inform you that your manuscript has been judged scientifically suitable for publication and will be formally accepted for publication once it meets all outstanding technical requirements.

Kind regards,

Shazia Rehman, Ph.D.

Academic Editor

PLOS ONE
---

## [Editor Report · Acceptance letter]

1 Mar 2024

PONE-D-23-39575R2 

PLOS ONE

Dear Dr. Jayathilaka, 

I'm pleased to inform you that your manuscript has been deemed suitable for publication in PLOS ONE. Congratulations! Your manuscript is now being handed over to our production team.

Kind regards, 

on behalf of

Dr. Shazia Rehman 

Academic Editor

PLOS ONE